# Generative Spoken Language Model based on continuous word-sized audio tokens

**Robin Algayres**[1,3]**, Yossi Adi**[2,3]**, Tu Anh Nguyen**[3]**, Jade Copet**[3]**,**
**Gabriel Synnaeve**[3]**, Benoit Sagot**[1]**, Emmanuel Dupoux**[1,3]
[1]ENS, INRIA, INSERM, UPEC, PSL Research University
[2]The Hebrew University of Jerusalem
[3]Meta AI
{robinalgayres,benoitsagot}@inria.fr
{adiyoss,edpx}@meta.com

## Abstract

In NLP, text language models based on words or subwords are known to outperform their character-based counterparts. Yet, in the speech community, the standard input of spoken LMs are 20ms or 40ms-long discrete units (shorter than a phoneme). Taking inspiration from word-based LM, we introduce a Generative Spoken Language Model (GSLM) based on word-size continuous-valued audio embeddings that can generate diverse and expressive language output. This is obtained by replacing lookup table for lexical types with a Lexical Embedding function, the cross entropy loss by a contrastive loss, and multinomial sampling by k-NN sampling. The resulting model is the first generative language model based on word-size continuous embeddings. Its performance is on par with discrete unit GSLMs regarding generation quality as measured by automatic metrics and subjective human judgements. Moreover, it is five times more memory efficient thanks to its large 200ms units. In addition, the embeddings before and after the Lexical Embedder are phonetically and semantically interpretable. [1]

## 1 Introduction

Recent work has opened up the possibility of learning generative language models directly from the raw audio signals, without using either text or Automatic Speech Recognition (ASR) (Lakhotia et al., 2021; Kharitonov et al., 2021b; Nguyen et al., 2022b; Borsos et al., 2022). The basic idea of these model is to rely on traditional text-based language models (LM), but replace the text input with some other discrete tokens directly learned from audio in an unsupervised fashion. The advantage of learning units from speech instead of relying on ASR is that this procedure can capture non-verbal vocalizations (like laughter) or intonation and rhythm, which are typically not transcribed, resulting in more expressive generations (Kreuk et al., 2021;

Kharitonov et al., 2021b). In addition, ASR may not be available in many languages that have insufficient textual resources and can make errors, which may then perturb the learning of the LM.

The problem of using self-discovered units, however, is that these units are typically very small, in fact, usually smaller than phonemes (Lakhotia et al., 2021; Borsos et al., 2022). We think that increasing the size of the units will favourably impact the semantic capabilities of a downstream spoken LM. This intuition comes from the NLP literature. Among others, Graves (2013); Mikolov et al. (2011); Bojanowski et al. (2015); Nguyen et al. (2022a) have shown a performance gap between character-based LM and word-based LM. The main reason is that at the level of characters, it is difficult for a text LM to extract long-range syntactic and semantic relationships. This is one of the reasons why recent state-of-the-art text-based LM (Radford et al., 2019) typically use a tokenizer representing word or subword units (Byte Pair Encoding (Gage, 1994), WordPiece (Wu et al., 2016), Unigram (Kudo, 2018)). Another advantage of large units is to save GPU memory at training time that enables to use both larger batches and longer sequences.

In speech, building the equivalent of a text-based tokenizer is hampered by two difficulties. First, the *boundary problem* is that contrary to text in most orthographic systems, speech does not have spaces and punctuation to delimit between word units. Finding word boundaries from raw audio is itself a difficult challenge (Dunbar et al., 2022a). Second, the *clustering problem*, is that even if boundaries were available, the clustering of speech fragments is challenging because the same word may surface in a variety of forms depending on speaker, accent, speech rate, etc. This problem may be even more difficult to solve than the first one (Dunbar et al., 2022a) because of the highly skewed distribution of word frequencies (Algayres et al., 2022b). Here,

---

[1]Audio examples are available at our website.

we investigate the possibility of building a *continuous tokenizer* that sidesteps these two problems by using tokens that have neither perfect boundaries nor require a clustering step. In Appendix B, we explain in more detail why we wish to avoid the clustering of speech fragments and what methods have been applied to tackle this problem so far.

Having a continuous tokenizer instead of a discrete one results in drastic changes from the point of view of the downstream LM. With a discrete tokenizer, one can define a finite list of tokens over which the LM can learn a lookup embedding table at the input of the model and use a softmax layer at the output of the model. The softmax is used in training mode to compute the loss function through a cross-entropy with the target token and at inference time to sample sentences. With continuous representations, the list of tokens is unbounded, making these computations intractable. We tackle this problem with a *Lexical Embedder*, a semi-learnable function that maps continuous tokens to a practically infinite list of embeddings.

The key question addressed in this paper is whether it is possible to generate speech using large (word-size) continuous units instead of short discrete ones. Our major technical contribution is to replace the three standard elements of a text-based LM (lookup table, cross-entropy loss function, multinomial sampling) with elements adapted to a virtually infinite list of continuous embeddings. We show that with these changes, it is possible to generate speech of the same quality as discrete unit models. This is interesting because our units are 200ms long which amounts to a 5-time memory reduction compared to regular discrete units (Lakhotia et al., 2021; Borsos et al., 2022), opening up the possibility to train spoken LMs on longer speech sequences. In addition, our model builds interpretable representations thanks to the Lexical Embedder which learns a mapping between an acoustic space, with phonetic properties, to a lexical space, with semantic and syntactic properties. We call the resulting model tGSLM (token-based GSLM).

## 2  Related work

**Unsupervised speech representations** like CPC, Wav2vec2.0 and HuBERT (van den Oord et al., 2018; Baevski et al., 2020; Hsu et al., 2021) are fixed-size representations (10 to 20ms long) that outperform traditional features, like mel-filterbanks and MFCCs, in many applications (Yang et al., 2021). In parallel to these works, there is a growing literature on variable-length acoustic encoding called speech sequence embeddings (SSE) (Peng et al., 2020; Algayres et al., 2022a; Jacobs et al., 2021; Kamper, 2018; Settle and Livescu, 2016). SSE models take a sequence of speech of any length and return a fixed-size vector. These models encode speech by maximizing phonetic information while minimizing speaker identity and recording conditions. SSEs are used for spoken term discovery (Thual et al., 2018), speech segmentation into phones or words (Kamper, 2022; Algayres et al., 2022b) but also as input to a BERT model (Algayres et al., 2022b) for spoken language modelling.

**Speech generation** is often performed with a neural vocoder conditioned on mel-filterbanks (van den Oord et al., 2016; Kumar et al., 2019; Kong et al., 2020; Prenger et al., 2018). In a text-to-speech pipeline, the mel-filterbanks are obtained with another neural network, which is conditioned on text (Ping et al., 2017; Shen et al., 2018). In the next step, the mel-filterbanks are decoded into natural-sounding speech by a neural vocoder (van den Oord et al., 2016; Kumar et al., 2019; Kong et al., 2020; Prenger et al., 2018). For the Zerospeech Challenge 2019, Dunbar et al. (2019) proposed to remove text and replace it with unsupervised discrete units. This challenge has fueled a large body of works on learning low bitrate speech representations for speech compression, voice conversion and spoken language modelling (Chen and Hain, 2020; Liu et al., 2019; Feng et al., 2019; Baevski et al., 2019; Tjandra et al., 2019; Kharitonov et al., 2021b; Lakhotia et al., 2021; Nguyen et al., 2020). For evaluation, the Zero-Resource challenge used bitrate and human evaluation.

**Spoken Language Model** are neural networks trained to predict missing parts of a spoken sentence with predictive or contrastive losses. GSLM (Lakhotia et al., 2021) is the first spoken LM able to generate expressive and consistent spoken sentences in a pure textless fashion. It uses a causal transformer LM trained with NLL loss on sequences of discrete units obtained with a $k$-means clustering (with $k$=100) of HuBERT frames. Once trained, GSLM can generate a sequence of discrete units by multinomial sampling that is de-

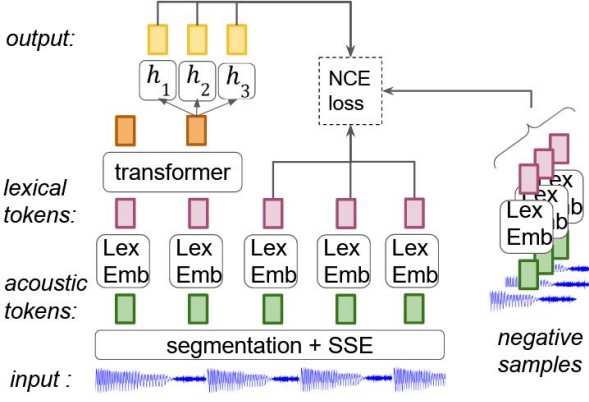

Figure 1: Speech is encoded into Wav2vec2.0 frames and segmented into chunks. These latter are converted into acoustic tokens with an SSE model and turned into lexical tokens by applying the function *LexEmb*. Finally, lexical tokens are fed to a causal transformer LM which attempts to predict the first, second, and third following tokens using parallel output heads. The acoustic tokens are pre-extracted before training the learnable modules (*LexEmb*, the transformer and the final fully connected layers) with the NCE loss. The negative samples are chosen randomly from other utterances of the same speaker.

coded into speech with a separate vocoder. Specifically, the sampled HuBERT units are mapped to mel-filterbanks with Tacotron2.0 and decoded into speech with *WaveGlow* (Prenger et al., 2018), a neural vocoder. Lakhotia et al. (2021) also provides a way to evaluate their spoken LM using an ASR to transcribe their spoken generations and an external LM to compute the perplexity of the resulting transcriptions. In addition, the Zerospeech Challenge 2021 (Nguyen et al., 2020) designed a set of zero-shot metrics to probe what spoken LMs learn. A recent paper (Borsos et al., 2022), audioLM, came to our attention, which we did not have the time to include in our experiments. AudioLM works similarly to GSLM yet with the ability to generate speech that preserves the identity of the speaker. In another line of work, Algayres et al. (2022b) trained a BERT model with a contrastive loss function on sentences represented as a series of SSEs. They showed the resulting BERT is able to model semantics and syntax. This work suggests that discrete tokenizers and the NLL loss are not necessary to tackle language modelling on speech. We take inspiration on their work to design our approach.

# 3 Approach

## 3.1 tGSLM: training

The general structure of tGSLM is presented in Figure 1. It is composed of an **encoder** which segments the input speech into sequences of possibly varying size, and computes a fixed-sized Speech Sequence Embedding (SSE), which we call acoustic tokens (Section 3.1.1). These tokens are turned into lexical tokens through a learnable **Lexical Embedder** (Section 3.1.2), and fed into a causal **Language Model** that has been modified to deal with continuous inputs (Section 3.1.3).

### 3.1.1 Acoustic tokens

In Figure 1, a speech sequence, $S$, is turned into $n$ acoustic tokens, $(a_0, ..., a_n)$, after applying speech segmentation and an SSE model.

Speech segmentation consists in finding word boundaries in a speech sentence (Algayres et al., 2022b; Kamper, 2022; Kreuk et al., 2020). In this work, we rely on a naive method by placing a boundary every 200 ms, regardless of the content of the speech signal. In the Appendix A.1, we show that this method leads to better results than recent, more complex speech segmentation systems.

The acoustic tokens $(a_i)_{i \leq n}$ are built by first encoding the speech sentence $S$ into a series of $n'$ frames $(f_i)_{i \leq n'}$ with the 8th layer of Wav2vec2.0 Base from Baevski et al. (2020). For any two boundaries $(k, l)$, $a_i = SSE([f_k, ..., f_l])$ where $SSE$ is a self-supervised system from Algayres et al. (2022a) trained with contrastive learning. This model has state-of-the-art performances on phonetic representation of pre-segmented words as measured by the Mean-Average-Precision metric. The acoustic tokens are extracted in a preprocessing step and stored before the training of the subsequent LM.

### 3.1.2 Lexical tokens

In a text-based transformer LM, there is often an embedding lookup table before the transformer, that has the size of the vocabulary and that maps discrete word tokens to lexical tokens (Vaswani et al., 2017). These lexical tokens, also known as word embeddings (Mikolov et al., 2013), learn during training semantic and syntactic properties that have been studied extensively in the NLP literature. In our case, the situation is different. First, instead of discrete word tokens, our LM takes as input continuous acoustic tokens which latent vo-

cabulary size is unknown. Second, the mapping between acoustic and lexical space cannot be linear, as two speech segments may sound the same, i.e. be close in the acoustic space, while being semantically/syntactically different, i.e. far in the lexical space. This highly non-linear function between acoustic and lexical space is learned by our lexical embedder: $LexEmb = L \circ q$ function. $L$ is a stack of non-linear fully connected layers learned jointly with the LM. $q$ is an information bottleneck quantization function that we had to introduce to minimize the presence of low-level non-linguistic acoustic information. For a speech sequence $S$ composed of $n$ acoustic tokens $(a_i)_{i \leq n}$, we note the sequence of lexical tokens $(l_i)_{i \leq n}$ such as $\forall i \leq n$, $l_i = LexEmb(a_i)$.

To understand why we need $q$, we have to go back to the LexEmb function input: the acoustic tokens. The acoustic tokens are derived from Wav2vec2.0, which is a transformer architecture whose attention mechanism covers the whole sentence. Each wav2vec2 frame, therefore, contains potential information about relative positions (through the transformer's positional embeddings), adjacent acoustic materials (through self-attention) or global properties like speaker. What we've found in preliminary experiments is that this information may leak into the acoustic tokens and be amplified by the prediction or contrastive loss of the downstream causal LM. Fortunately, it turns out that this information has low variance and can be partially removed by slightly degrading the quality of the acoustic tokens. The degradation of the acoustic tokens is the role of the function $q$. $q$ is composed of a PCA reduction and a quantization step that we call *d-k-means*, which stands for per-dimension k-means. Specifically, given a speech database that has been segmented and encoded into $N$ acoustic tokens, $(a_i)_{i \leq N}$, we reduce their dimensions to $d$ with a PCA. Then, we train $d$ different k-means, one for each dimension of the PCA. In other words, for each $j \leq d$, we train a k-means on $(PCA(a_i)[j])_{i \leq N}$. We chose the number of centroids per k-means to be proportional to the explained variance of each of the PCA dimensions. Once the k-means are trained, each dimension of each acoustic token is mapped to its cluster id. Finally, the cluster ids are turned into one-hot vectors and concatenated into one vector (see Appendix A.2 for more detailed explanations). d-k-means is inspired from multi-stage vector quantizer (VQ)

(Vasuki and Vanathi, 2006) where several VQ codebooks are learned in parallel as in Baevski et al. (2020); Zeghidour et al. (2021). The PCA and the d-k-means are trained over the whole training set as a preprocessing step, before the transformer LM. We ablate the use of $q$ in Appendix A.2 and show that it is necessary for the LM to generate sentences[2].

### 3.1.3 Causal language model

The LM is a standard causal transformer with two modifications: the loss function and the prediction heads. First, in a standard LM, the number of possible types is fixed beforehand and remains tractable even for a very large corpus (10k to 100k). Here, because the number of different lexical tokens is virtually infinite, we cannot use a standard softmax and cross-entropy loss. We first tried a simple L2 reconstruction loss with an additional decoder but it did not work for us in practice. Instead, we use a contrastive loss: the Noice Contrastive Estimation (NCE) loss (Gutmann and Hyvärinen, 2010). This loss works by maximizing the similarity between a pair of positive samples while minimizing the similarity between the positive samples and various negative samples. However, even though the SSE model from Algayres et al. (2022a) has learned to be speaker invariant, there is still a lot of speaker-related information encoded into the acoustic tokens. This is a problem already encountered in Algayres et al. (2022a); van den Oord et al. (2018) that is dealt with by sampling the negative tokens from the same speaker as the positive tokens.

Second, in a standard LM, the output head typically predicts the next word. However, in the case of speech, the boundary between individual phonemes is blurred by coarticulation. It is therefore easy to predict the next word by just attending to very local acoustic information at the end of the last word (something impossible to do with characters which are sequentially disentangled). We, therefore, introduce three prediction heads (three linear fully connected layers: $h_1, h_2, h_3$) which do not only predict the first next token, but also the second and third as they cannot be co-articulated with the last token encoded by the LM. These prediction layers are trained jointly with the LM. We

---

[2]Due to this quantization step, the resulting vectors (PCA+ d-k-means) could in principle be mapped to a finite dictionary of tokens, but, in practice, there is little or no collision and the number of classes remains identical to the number of tokens, i.e., way too high to apply a softmax.

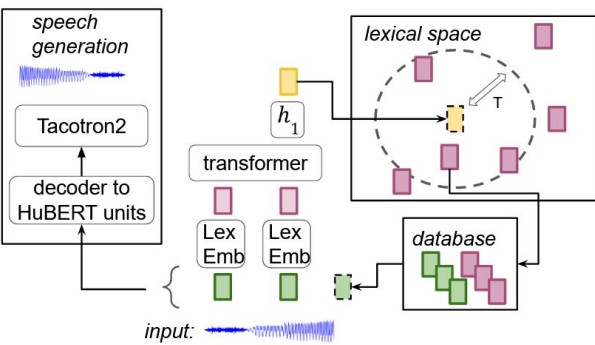

Figure 2: Our sampling procedure. Given a list of audio files unseen during training, $N$ random speech segments are stored in their acoustic and lexical forms: $(a_i, l_i)_{i \leq N}$. In addition, a *lexical space* is created by indexing $(l_i)_{i \leq N}$ into a k-NN graph. Given a speech prompt, segmented and encoded into $(a_0, ..., a_t)$, we do a forward pass in tGSLM and search for the nearest neighbors of $h_1$ output in the lexical space. $l_{t+1}$ is sampled and its corresponding $a_{t+1}$ is appended to $(a_0, ..., a_t)$. When a final $a_T$ token is sampled, $(a_0, ..., a_T)$ is decoded into HuBERT units and speech is generated with Tacotron2.

justify the choice of three prediction heads with a grid-search available in Appendix Table 5.

## 3.2 tGSLM: generation

Once tGSLM training is done, we use it to generate spoken sentences. We do that in two steps: we generate a sequence of acoustic tokens (Section 3.2.1) and then decode this sequence into speech (Section 3.2.2).

### 3.2.1 Sampling

To generate a spoken sentence, we take inspiration of the popular top-k sampling method used in NLP to generate text sentences. This method requires sampling series of word tokens by sampling among the most probable word types. In our case, we do not have access to types so we are going to sample among the most probable lexical tokens. Our sampling method is summarized in Figure 2. We start by collecting a few dozen hours of speech that have not been seen during tGSLM training. The utterances are segmented and encoded into $N$ speech segments and stored in their acoustic and lexical forms: $(a_i, l_i)_{i \leq N}$. Using the FAISS library (Johnson et al., 2017), we index $(l_i)_{i \leq N}$ into a k-NN graph called the lexical space. Given a prompt of $t$ acoustic tokens $(a_0, ..., a_t)$, we do a forward pass into tGSLM. Then, we compute the cosine similarity of $h_1$ output and its $k$ closest neighbours in the lexical space. We apply a softmax on the vector of cosine similarities and treat it as a multinomial distribution to sample one element: $l_{t+1}$. The softmax function contains a temperature parameter that controls the range of the sampling area. The acoustic tokens $a_{t+1}$ that correspond $l_{t+1}$ is retrieved from the stored database and appended to $(a_0, ..., a_t)$. Once the desired length is reached, the sequence of acoustic tokens is decoded into a spoken sentence as explained in the next section.

### 3.2.2 Speech generation

Lakhotia et al. (2021); Kharitonov et al. (2022) trained a Tacotron2.0 decoder (Shen et al., 2018) to map deduplicated HuBERT units into mel filterbanks. Then, speech is generated from the mel filterbanks by a *WaveGlow* vocoder (Prenger et al., 2018). In order to make use of this pre-trained Tacotron2.0 decoder, we trained an encoder-decoder transformer model to map series of acoustic tokens to series of HuBERT units. During training, the encoder computes an attention over a series of acoustic tokens while the decoder predicts HuBERT units auto-regressively. At inference, given a series of acoustic tokens, a corresponding sequence of HuBERT units is obtained by taking the argmax of the decoder softmax function. Finally, the HuBERT units are given as input to the pre-trained Tacotron2.0 to be decoded into spoken utterances.

## 4 Evaluation and datasets

### 4.1 Datasets and settings

LJ Speech (LJ), LibriSpeech (LS), Libri-light 6k clean (LL6k-clean) are three corpora of studio recordings of read English of respectively 24, 1k and 6k hours (Ito and Johnson, 2017; Panayotov et al., 2015; Rivière and Dupoux, 2021). These corpora are used to train the different parts of the pipeline. The training details and specific model architectures can be found in Appendix Section A.3.

### 4.2 Generation metrics

**Perplexity** (PPX) is a text-based metrics used by Lakhotia et al. (2021) to evaluate the overall quality of generated spoken sentences. The authors propose to transcribe the spoken generations with an external ASR system and to compute the mean perplexity score over batches of transcribed speech

with an external transformer LM[3]. The spoken generation process is guided by a temperature parameter that controls how diverse generated sentences are. The diversity of a batch of sentences can be computed as in Lakhotia et al. (2021) with the VERT score that is an average of self-BLEU (Zhu et al., 2018) and auto-BLEU (Lakhotia et al., 2021) scores. Typically, low temperatures produce high diversity and low perplexity, whereas high temperatures produce low diversity and high perplexity.

Finally, the perplexity of spoken generation is a metric that presents a high variance, therefore, as a compromise between acceptable generation time and low variance, we compute perplexity over batches of 100 generated utterances whose transcriptions are each exactly 30 words (around 10 seconds of audio).

**Subjective judgements** are computed with the meaningful Mean Opinion Scores (MMOS) in which human raters were asked to evaluate how natural (considering both grammar and meaning) a given spoken generation is. For both subjective tests, raters evaluate the samples on a scale of 1-5 with an increment of 1. We follow the method from Lakhotia et al. (2021) where they evaluated 100 samples from each of the evaluated methods while enforcing at least 15 raters for each sample. The CrowdMOS package (Ribeiro et al., 2011) was used with the recommended recipes for detecting and discarding inaccurate scores. As for the perplexity measures, the sentences are generated without conditioning on a prompt.

### 4.3 Zero-shot metrics

$sWUGGY$ **and** $sBLIMP$ are zero-shot tasks to evaluate spoken language models introduced in the Zerospeech Challenge 2021 (Nguyen et al., 2020):. These metrics are inspired by psycholinguistics and are used for interpreting what spoken LM learns. $sWUGGY$ is a list of pairs of word/non-word synthesized with the Google TTS API and filtered for the words that are in the LibriSpeech training set. $sBLIMP$ is a list of pairs of syntactically correct/incorrect synthesized sentences. Both $sWUGGY$ and $sBLIMP$ require the spoken LM to attribute a higher probability to

the correct element in each pair. Probabilities are computed by applying the spoken LM training loss directly on the test items.

$ABX_{sem}$ **and** $ABX_{POS}$ are additional zero-shot tasks introduced in Algayres et al. (2022b) to evaluate the semantic encoding and Part-Of-Speech (POS) tagging, this time not based on probabilities but on distances between embeddings. An ABX task is a list of triplets $A$,$B$ and $X$ where $A$ and $B$ belong to the same category and $X$ is a distractor. The task is to encode the triplet with a distance $d$ and show that $d(A, B) < d(A, X)$. In this case, $A$,$B$, and $X$ are spoken words given in the context of a sentence. For $ABX_{sem}$, A and B are close semantically, and X is random. For $ABX_{POS}$ A and B share the same POS tag, and X has different POS tags.

**Normalised Edit Distance** (NED) introduced in Versteegh et al. (2016) is a term discovery task that consists in finding clusters or pairs of speech segments from unsegmented audio that have the same phonetic transcription. For each discovered pair, the NED is computed as the edit distance normalized by the length of the longest item in the pair. As for ABX tasks, the NED is also based on the distance between embeddings. To compute a NED score, we take inspiration of the procedure introduced in Thual et al. (2018). Given a segmentation of the LibriSpeech dev-clean subset, all speech segments are embedded into fixed-size vectors. With a k-NN, we search for the pairs of closest embeddings and sort them by cosine similarity. Starting from the higher similarities, we retrieve as much pair as necessary to cover the whole dev-clean set. With the phoneme-level transcription of the dev-clean set, all pairs can be transcribed into series of phonemes. The final NED score is obtained by averaging the NED over all pairs of transcriptions. NED and ABX tasks both rely on embeddings that can be extracted at any level of a multi-layer neural model.

## 5 Results

### 5.1 Generation performances

#### 5.1.1 Perplexity and diversity

Figure 3 provides a comparison of the original discrete unit-based GSLM with two versions of our continuous unit model: 200ms-tGSLM, trained on speech segmented every 200ms and gold-tGSLM, trained on speech segmented on the true word boundaries. GSLM and 200ms-tGSLM are trained

---

[3]ASR transcripts are obtained with a pretrained large Wav2Vec 2.0 model, trained on LibriSpeech-960h combined with a standard KenLM 4-gram LM. The external LM used for perplexity is trained on the English NewsCrawl dataset and accessible at `https://github.com/facebookresearch/fairseq/tree/main/examples/language_model`

| | Zero-shot metrics↑ | | | | Generation PPX↓ | | Generation MMOS↑ | |
|---|---|---|---|---|---|---|---|---|
| | sWUGGY | SBLIMP | $ABX_{sem}$ | $ABX_{POS}$ | LS-VERT | LJ-VERT | LS-VERT | LJ-VERT |
| GSLM | **70.36** | **56.31** | 55.85 | 59.03 | **503.25+-12.3** | 387.45+-11.2 | 3.76 +- 0.035 | 3.78 +- 0.023 |
| 200ms-tGSLM | 68.53 | 55.31 | **55.89** | **60.3** | 532.87+-10.1 | **356.24+-15.7** | **4.09 +- 0.016** | **4.04 +- 0.020** |
| *gold-tGSLM* | *86.37* | *-†* | *65.6* | *75.59* | *361.84+-20.1** | *255.32+-14.2** | *n/a* | *n/a* |
| *character-gold* | *n/a* | *n/a* | *n/a* | *n/a* | *180.2* | *142.6* | *4.12 +- 0.016* | *4.11 +- 0.023* |

Table 1: Results on zero-shots and generation tasks for 200ms-tGSLM and GSLM, trained on LL6k-clean, and gold-tGSLM, trained on LibriSpeech. ABX is computed on tGSLM lexical tokens and on GSLM 9th layer. The last line is a topline that is composed of true sentences from both LibriSpeech and LJ. *: those scores are obtained without using a speech decoder. † time-aligned word boundaries for sBLIMP are not available

.

Figure 3: PPX and VERT scores for GSLM, 200ms-tGSLM and gold-tGSLM. Each dot is obtained by generating sentences with a fixed temperature parameter. The curves are 3rd-degree polynomial interpolations of the dots. The green dashed lines are the oracle PPX/VERT obtained on the LibriSpeech and LJ corpus.

on LL6k-clean[4] while the topline, gold-tGSLM, is trained only on LibriSpeech corpus[5]. The dots in Figure 3 represent batches of generated sentences conditioned on different temperatures. Color curves are the 3rd-degree polynomial interpolation of the dots. In green dashed lines appear two VERT anchor points LJ-VERT(=0.113) and LS-VERT(=0.189). These points are the mean VERT scores obtained on batches of sentences from, respectively LJ and LibriSpeech datasets. The intersection of the dashed lines and the curves gives the scores PPX@LS-VERT and PPX@LJ-VERT that are reported in Table 1[6].

Regarding the perplexity scores from Table 1, compared to GSLM, 200ms-tGSLM is slightly better at LJ-VERT and slightly worse at LS-VERT. The measure of perplexities being very noisy, these scores show that both models have similar performances. Some examples of transcribed spoken generations are available in Appendix Tables 8,9 and 10.

The topline gold-tGSLM produces much lower perplexities than GSLM and 200ms-tGSLM. Yet, we have experienced a problem with the speech decoder (described in Section 3.2.2) of gold-tGSLM. The scores of our topline are obtained by retrieving the exact transcriptions of the sampled SSEs instead of decoding them with the speech decoder. We had to do this because our speech decoder makes a lot of decoding mistakes when it tries to decode SSEs of variable-size speech fragments. It seems to generate fully intelligible speech only when it is trained to decode SSEs of same-size speech chunks, as is the case for 200ms-tGSLM. We think this happened because, for a lack of time and resources, we chose a poor decoding strategy (decoder from SSEs to HuBERT frames and HuBERT frames to speech). In our future works, we will focus on training a model to decode the SSEs directly into speech, using, for instance, recent diffusion models or a Hi-Fi Gan (Polyak et al., 2021; Huang et al., 2022). As a consequence of the poor performances of our speech decoder, we have not been able to leverage recent progress in speech segmentation into words (Algayres et al., 2022b; Kamper, 2022; Peng and Harwath, 2023) that provide word boundaries more aligned with real words than our 200ms chunks. In Appendix A.1 are the results of our attempts at using speech segmentation systems.

---

[4]Training 200ms-tGSLM on Libri-light 60k (Kahn et al., 2019), a larger but noisier corpus, slightly undermined the performance.

[5]word boundaries cannot be computed for LL6k-clean because sentence-level speech and text alignments are missing

[6]For a given spoken LM, its PPX@LS-VERT score is the perplexity score obtained by that spoken LM when conditioned on a temperature that makes it generate spoken sentences with a VERT equal to the VERT of the LibriSpeech.

| models | tokens | NED $\downarrow$ | $ABX_{sem} \uparrow$ | $ABX_{POS} \uparrow$ |
|---|---|---|---|---|
| 200ms-tGSLM | acoustic | **34.51** | 50.14 | 49.87 |
| | lexical | 47.98 | **55.08** | **60.24** |
| gold-tGSLM | acoustic | **16.15** | 50.20 | 50.12 |
| | lexical | 22.70 | **65.60** | **75.59** |

Table 2: NED and ABX scores on acoustic and lexical tokens for 200ms-tGSLM and gold-tGSLM both trained on LibriSpeech. ABX and NED are computed on tGSLM lexical tokens

### 5.1.2 Subjective judgements

As for perplexity, we report in Table 1, the MMOS for batches of spoken generations that have a diversity score equal to the VERT of either LibriSpeech (MMOS@LS-VERT) or LJ (MMOS@LJ-VERT). In addition to 200ms-tGSLM and GSLM we evaluate a topline called *character-gold* that are speech utterances obtained with Text-To-Speech (Tacotron2.0 from Shen et al. (2017)) taking in input the transcriptions of LJ and LibriSpeech utterances. From Table 1, for the high-temperature regime that leads to diversity scores in the range of LJ and Librispeech, 200ms-tGSLM is slightly better than GSLM and gets close scores with the topline. MMOS scores are not available for gold-tGSLM has the speech decoder did not work properly. Nonetheless, our table of results does not show the performances of tGSLM in a much lower temperature regime. When conditioned on very low temperature, GSLM can generate very simple and intelligible sentences, whereas 200ms-tGSLM start to produce gibberish. Therefore, both models have their strengths and weaknesses.

### 5.2 Zero-shot performances

To complete our analysis, we provide in Table 1, performances on the zero-shot tasks scores that are comparable for GSLM and 200ms-tGSLM. GSLM has a little advantage on $sWUGGY$ and $sBLIMP$ and an 200ms-tGSLM a slight advantage on $ABX_{sem}$ and $ABX_{POS}$. The topline gold-tGSLM, once again gets much stronger results. ABX scores are obtained, for GSLM at the 9th layer of the transformer and for tGSLM with the lexical tokens.

### 5.3 Interpretability

In order to analyze what is learned by $LexEmb$ we measure the ABX and NED of lexical tokens and acoustic tokens. In Table 2, the ABX scores show that the acoustic tokens are at chance level on semantic and syntactic encoding. After the $LexEmb$ function, the lexical tokens lose a bit of their phonetic encoding (NED increases) but gain the ability to represent semantics and syntax. However, the NED is not at chance level, meaning that a bit of acoustic information has leaked into the lexical tokens. To visualize the difference between acoustic and lexical spaces, we provide t-SNE maps in Appendix Section A.4.

### 5.4 Memory consumption

GSLM model (Lakhotia et al., 2021) and 200ms-tGSLM use the same transformer LM but with different types of inputs. Compared to the 200ms-long units of our model, GSLM is trained on discrete units that are 40ms long on average (when contiguous duplicates are removed). Therefore, we expected our model to be more memory efficient than GSLM[7] which can be observed by the maximal batch size that both models can handle. Indeed, on the one hand, we managed to train GSLM with 34 60-seconds-long sentences on a 32G V100 GPU without OOM error. On the other hand, 200ms-tGSLM can fit as many as 162 sentences, which shows almost a 5-time reduction ($\approx 4.76$) of memory use.

Training spoken LMs on long sequences of audio will become necessary in order to learn long-term semantic relations. The usage of very short acoustic units can become a bottleneck which our method helps to alleviate. To complete our analysis, we provide in Appendix A.5 a theoretical analysis of memory reduction.

## 6 Conclusion

We introduced a generative spoken LM based on continuous word-sized acoustic tokens. Our model is able to generate speech with the same level of diversity and accuracy as a model based on discrete units. This shows that building a lexicon of types is not necessary for spoken language modelling, which is encouraging considering the difficulty of clustering large segments of speech without degrading the representation (see Appendix B). In addition, this performance was obtained with segments that were not very well aligned with word boundaries (200ms segments). The good result obtained with gold word boundaries indicates that there is room for improvement by using segments

---

[7]The acoustic tokens that are the input of 200ms-tGSLM are extracted in a preprocessing step. They do not impact memory usage at training time.

better aligned with word boundaries and of course a better speech decoder. Further work is also needed to better limit the leakage of low-level acoustic information into the LM through continuous units, which our analysis has shown is detrimental to the performance of the generative model (see also Nguyen et al. (2022c)). Finally, the fact that the units are about 5 times larger than standard GSLM units aligns with the NLP literature that is in favour of word-based LMs. It opens the possibility to fit larger spans of audio in GPUs and capture long-distance relationships.

## 7 Limitations

Our method has some limitations that range from GPU consumption, potential overfitting on the English language and sub-optimal decoding method. First, tGSLM is trained on 32 Nvidia V100-32Go GPUs for 30 hours. Due to the several modules at work in tGSLM (SSE model, LexEmb function, transformer decoder and seq2seq decoder), a large grid-search on hyper-parameters has been necessary which makes this work quite resource-consuming. Secondly, during the grid-search we chose hyper-parameters to optimize the semantic and syntactic ABX scores on English. By doing so, we might have overfitted the English language and made tGSLM specifically good at generating English speech. Further analysis is required to see if our method generalizes well to syntactically and morphologically different languages, like French or Mandarin. Finally, our decoding method is based on a seq2seq transformer that produces HuBERT frames which are decoded into speech with a combination of Tacotron2.0 and WaveGlow. We chose that method as this later speech synthesiser comes pre-trained in the *textlesslib* Python library (Kharitonov et al., 2022). Yet, recent work on *textless* speech synthesis Kreuk et al. (2021); Kharitonov et al. (2021a) skip the spectrogram prediction of Tacotron2.0 and directly train a Hifi-Gan model to generate speech from HuBERT units. This latter model shows close to human-level performances. We leave the use of Hifi-Gan instead of Tacotron2.0 for future works on tGSLM.

## 8 Ethical statement

tGSLM is a LM that learns to generate speech sentences by predicting its training data. Therefore, tGSLM inherits from ethical concerns associated with text-based LM, speech encoders and speech synthesizers. It is of paramount importance to safeguard against these issues.

First, generative text-based LMs are known to repeat stereotypes and biases that belong to the training corpus which can cause a series of harms Chowdhery et al. (2022); Bender et al. (2021). One way to mitigate this is to apply post-processing on generated sentences to detect harmful content. Yet, from what we have heard, tGSLM still struggles to generate sentences that fully make sense, so we do not think that post-processing is required at the moment.

Second, if tGSLM is used to continue a speech prompt, the continuation might be inconsistent for accents of underrepresented groups in the training data. Indeed, speech systems are known to encode poorly accents and dialects out of the training distribution (Riviere et al., 2021).

Finally, tGSLM continuations will not preserve any regional accentuation from the prompt, as our model only generates speech in the voice of the single speaker of the LJ dataset.

## Acknowledgements

This work was funded in part, to the authors in their academic capacities, by the Agence Nationale pour la Recherche (ANR-17-EURE-0017 Frontcog, ANR-10-IDEX-0001-02 PSL*, ANR-19-P3IA-0001 PRAIRIE 3IA Institute), CIFAR (Learning in Machines and Brains) and Meta AI Research (Research Grant). This work was performed using HPC resources from GENCI-IDRIS (Grant 2021-[AD011011217]).

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

# A  Supplementary materials

## A.1  Speech segmentation

To study the impact of speech segmentation on tGSLM, we trained this model on LibriSpeech with two extra segmentation methods: SylSeg (Räsänen et al., 2018), and DP-Parse (Algayres et al., 2022b)[8]. Sylseg segments speech into syllable-like units, using damped oscillators that exploit rhythmic cues of syllabic structure in speech. DP-Parse (Algayres et al., 2022b) segments speech into word-like units with state-of-the-art performances. This model adapts a non-parametric Bayesian model for text segmentation (Goldwater et al., 2009)

---

[8]We did not train these models on LL6k-clean because DP-Parse is hard to scale to large datasets.

to speech. Table 3 shows generation and zero-shot scores. Overall, regarding speech generation, 200ms-tGSLM outperforms sylseg-tGSLM, dpparse-tGSLM and also GSLM. For zero-shot tasks, once again, all models score similarly. ABX scores are again obtained for GSLM with embeddings extracted from the 9th layer of the transformer and for tGSLM from the lexical tokens.

Even though true word boundaries strongly benefit tGSLM, using unsupervised speech segmentation methods did not prove beneficial. We think this is due to the low performances of state-of-the-art speech segmentation systems. These latter are only marginally better than random segmentations and lag largely behind text segmentation performances (Dunbar et al., 2022b; Algayres et al., 2022b). This result suggests that progress is needed in unsupervised speech segmentation to be able to combine segmented units into intelligible speech. After all, the best segmentation method that we works for us is the 200ms method. We have also experimented with other durations as 120ms,280ms and 360ms. We chose to go on with 200ms based on a compromise between maximal duration and maximal zero-shot task performances. These scores can be found in Appendix Table 4.

## A.2  Discussion on $q$

### A.2.1  Mathematical details on $q$

Let us now derive $q$ computation. Given a training corpus, that is segmented and encoded into a collection of acoustic tokens $(a_i)_{i \leq N}$. A PCA is trained on $(a_i)_{i \leq N}$ and the $d$ first dimensions are kept, let us write $(a'_i)_{i \leq N}$ the resulting vectors and $(v_0, ..., v_{d'})$ the explained variance of each PCA dimensions. Then, we train $d$ separate k-means on each dimension of the PCA. The number of cluster per k-means is computed as $(\left\lceil K \frac{v_0}{v_0} \right\rceil, , , ..., \left\lceil K \frac{v_{d'}}{v_0} \right\rceil)$. The values of $d$ and $K$ were set to maximize the scores at the zero-shot tasks. Once the k-means are trained, the centroids are stored in $d$ dictionaries $(k_0, ..., k_d)$. For any $i \leq N$, we compute $q(a_i)$ by assigning $\forall j \leq d$, $q(a_i)[j]$ to its closest centroids in $k_j$. Finally, cluster ids are turned into one-hot vectors and concatenated into a single vector. The following operations

| | WUGGY↑ | SBLIMP↑ | $ABX_{sem}$↑ | $ABX_{POS}$↑ | PPX@LS-VERT↓ | PPX@LJ-VERT↓ |
|---|---|---|---|---|---|---|
| GSLM | **65.85** | **54.35** | 55.18 | **61.61** | 664.23 | 497.65 |
| sylseg-tGSLM | 64.39 | 53.21 | 54.64 | 60.01 | 634.34 | 505.87 |
| dpparse-tGSLM | 65.54 | 53.82 | **55.6** | 58.65 | 634.34 | 505.87 |
| 200ms-tGSLM | 63.15 | 53.34 | 55.08 | 60.24 | **610.32** | **490.32** |

Table 3: Results on zero-shot and generation tasks for GSLM and for tGSLM on three different speech segmentation methods. Models are all trained on LibriSpeech. ABX is computed on tGSLM lexical tokens and on GSLM 9th layer

| models | $sWUGGY$↑ | $sBLIMP$↑ | $ABX_{sem}$↑ | $ABX_{POS}$↑ | $average$↑ |
|---|---|---|---|---|---|
| 120ms-tGSLM | 61.55 | 51.86 | 54.74 | 60.12 | 57.32 |
| 200ms-tGSLM | 63.15 | 53.34 | 55.08 | 60.24 | 57.95 |
| 280ms-tGSLM | 61.89 | 51.64 | 52.8 | 56.28 | 55.65 |
| 360ms-tGSLM | 60.18 | 51.29 | 52.18 | 55.45 | 54.75 |

Table 4: Zero-shot tasks computed on tGSLM trained on LibriSpeech for different unit durations

sum up the process.

$$\forall i \leq n, q(a_i) \leftarrow \begin{pmatrix} \underset{j \leq K}{\mathrm{argmax}}(a_i[0] - k_0[j]) \\ \underset{j \leq \left\lceil K\frac{v_1}{v_0} \right\rceil}{\mathrm{argmax}}(a_i[1] - k_1[j]) \\ \vdots \\ \underset{j \leq \left\lceil K\frac{v_d}{v_0} \right\rceil}{\mathrm{argmax}}(a_i[d] - k_d[j]) \end{pmatrix}$$

$$q(a_i) \leftarrow \begin{pmatrix} onehot(q(a_i[0])) \\ onehot(q(a_i[1])) \\ \vdots \\ onehot(q(a_i[d])) \end{pmatrix}$$

$$q(a_i) \leftarrow concatenate(q(a_i[0]), ..., q(a_i[d]))$$

### A.2.2 Ablation on $q$

The function $q$ introduced in Section 3.1.2, composed of a PCA and our d-k-means method, is ablated in Table 6. In all configurations, the embeddings right after the $LexEmb$ function are used to compute the ABX and NED scores. On the one hand, $q$ degrades the phonetic information in the lexical tokens (NED increases) and makes training harder (validation loss increases). On the other hand, $q$ maximizes semantic and syntactic information (ABX increases) as well as generation quality (PPX decreases). A $null$ value in Table 6 means that the model is not able to produce intelligible sentences with this setup. First, these experiments show the necessity of $q$ for the 200ms-tGSLM to generate spoken sentences. Second, the combination of these results reveals that $q$ prevents the model from converging quickly to a bad local minimum that hinders generalization.

It follows our intuition from Section 3.1.2: there seems to be a low-variance signal encoded in the

acoustic tokens that interfere with the semantic and syntactic modelling. In our opinion, this signal gives away both local information, direct right and left context due to coarticulation, and global sentence-level information (relative token position and speaker identity).

### A.2.3 On using MFCCs instead of Wav2vec2.0

One may say that if $q$ is used to mitigate the downsides of the attention mechanism of Wav2vec2.0, why not use more local features like MFCC or Mel-filterbanks? We argue that even though these latter features are still good for supervised tasks as ASR Radford et al. (2022), they are substantially outperformed by recent self-supervised speech models (Wav2vec2.0, CPC, HuBERT,...) at the tasks of zero-shot word discrimination (Algayres et al., 2022a; Van Staden and Kamper, 2020) and keyword spotting (Yang et al., 2021). To prove our point, we compare the performances of MFCCs compare to Wav2vec2.0 at the task of discriminating acoustic tokens. As a reminder, the acoustic tokens that we used in our model, are the output of an SSE model from Algayres et al. (2022a), pre-trained to embed variable-length sequences of Wav2vec2.0 frames into fixed-size vectors. The same kind of SSE model but pre-trained on MFCC frames is also provided by Algayres et al. (2022a). Let us segment the LibriSpeech corpus every 200ms and embed the speech segments with both SSE models so that we get two collections of acoustic tokens: $(a_i^{mfcc})_{i \leq N}$ and $(a_i^{w2v2})_{i \leq N}$. Let us also apply the $q$ function on Wav2vec2.0 acoustic tokens so that we get: $(q(a_i^{w2v2}))_{i \leq N}$. To measure performances, we use our $NED$ metric on the three collections of embeddings. From the

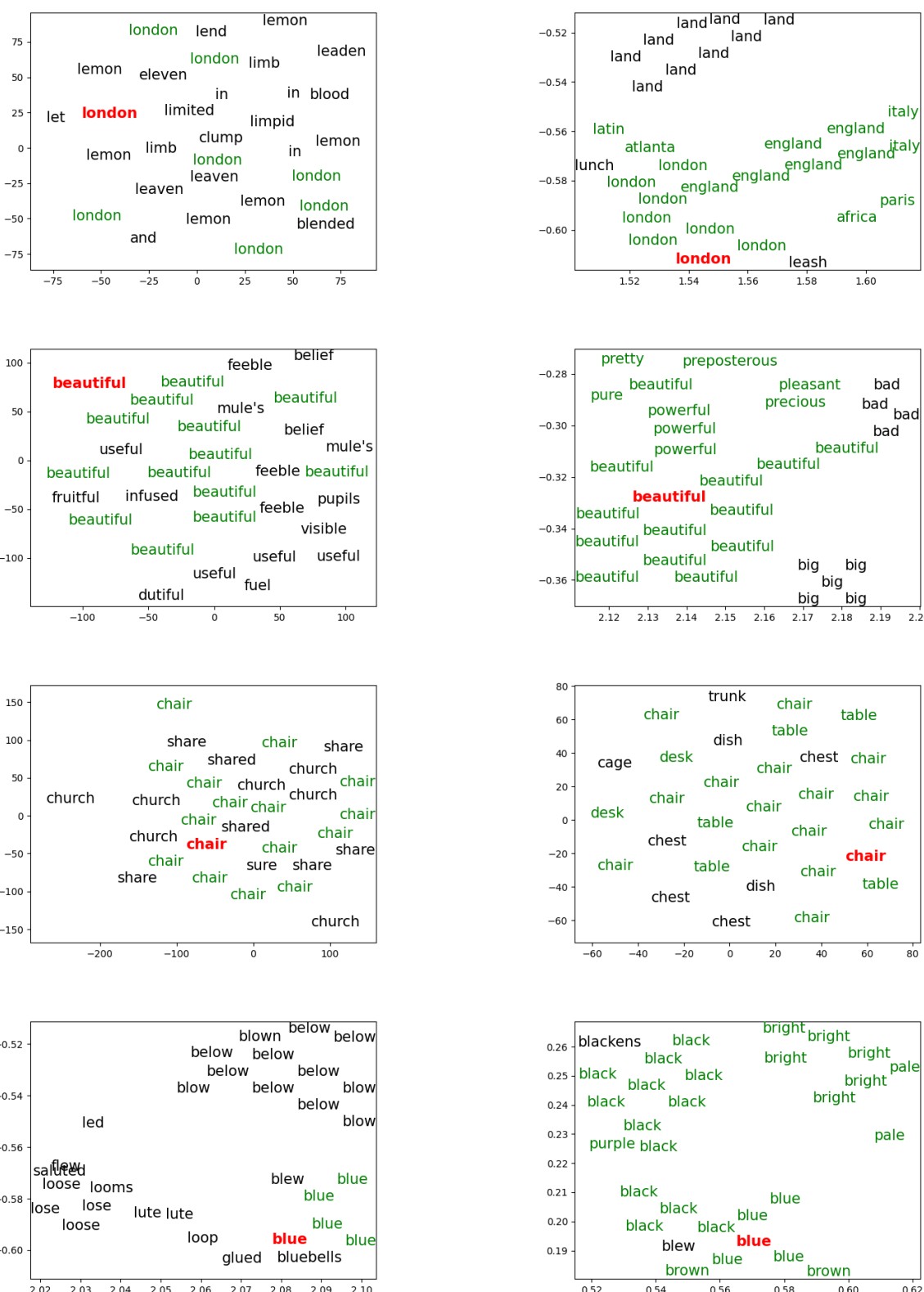

Figure 4: t-SNE representations of acoustic (left side) and lexical (right side) tokens. After training gold-tGSLM, all speech segments corresponding to word tokens in the LibriSpeech dev-clean subset are indexed into their acoustic or lexical form. By probing an acoustic or lexical token (appearing in red), we can have a look at their acoustic and lexical nearest neighbours. The neighbours that appear in green are those deemed as semantically related to the probe.

| models | $sWUGGY \uparrow$ | $sBLIMP \uparrow$ | $ABX_{sem} \uparrow$ | $ABX_{POS} \uparrow$ | $average \uparrow$ |
|---|---|---|---|---|---|
| next word | 61.57 | 52.08 | 51.48 | 53.84 | 54.75 |
| next two words | 63.02 | **53.48** | 54.79 | 58.01 | 57.35 |
| next three words | **63.15** | 53.34 | **55.08** | **60.24** | **57.95** |
| next four words | 62.25 | 53.1 | 54.43 | 58.81 | 57.14 |

Table 5: Zero-shot tasks for 200ms-tGSLM trained on LibriSpeech to predict the next one, two, three, or four words

| | PCA | d-k-means | **Valid loss**↓ | **NED**↓ | $ABX_{sem} \uparrow$ | $ABX_{POS} \uparrow$ | **PPX@LS-VERT**↓ | **PPX@LJ-VERT**↓ |
|---|---|---|---|---|---|---|---|---|
| 200ms-tGSLM | | | **2.51** | **35.21** | 53.87 | 58.40 | null | null |
| 200ms-tGSLM | ✓ | | 4.33 | 41.50 | 54.16 | 57.99 | 840.65 | null |
| 200ms-tGSLM | ✓ | ✓ | 6.21 | 44.32 | **55.08** | **60.24** | 610.32 | 490.32 |
| gold-tGSLM | | | **3.99** | **17.21** | 55.13 | 63.54 | 608.24 | 475.65 |
| gold-tGSLM | ✓ | | 6.20 | 21.87 | 58.59 | 67.71 | 432.78 | 384.57 |
| gold-tGSLM | ✓ | ✓ | 7.15 | 22.70 | **65.60** | **75.59** | **361.84** | **255.32** |

Table 6: Results on zero-shot and generation tasks for ablations of the PCA and d-k-means components of the $LexEmb$ function. Models are trained on LibriSpeech. ABX and NED are computed on tGSLM lexical tokens. $null$ means that no intelligible speech can be generated in this setting.

results Table 7, we see that Wav2vec2.0 leads to much better acoustic tokens than MFCCs. Moreover, even when $q$ is applied on Wav2vec2.0 acoustic tokens, the $NED$ score of $(q(a_i^{w2v2}))_{i \leq N}$ is still much lower than on $(a_i^{mfcc})_{i \leq N}$. This latter has a NED score of 65%, which means that two neighbouring MFCC acoustic tokens have on average less than half of their phonemes in common. For that reason, why we excluded MFCC from our experiments on speech generation.

| acoustic tokens | $NED \downarrow$ |
|---|---|
| $(a_i^{mfcc})_{i \leq N}$ | 65.56 |
| $(a_i^{w2v2})_{i \leq N}$ | 31.81 |
| $(q(a_i^{w2v2}))_{i \leq N}$ | 36.71 |

Table 7: NED scores of 200ms-long acoustic tokens built on MFCC: $(a_i^{mfcc})_{i \leq N}$, on Wav2vec2.0 frames: $(a_i^{w2v2})_{i \leq N}$, and finally on Wav2vec2.0 frames when $q$ is applied: $(q(a_i^{w2v2}))_{i \leq N}$
.

### A.3 Hyperparameters

**Wav2vec2.0 and SSE** are trained on the LibriSpeech corpus respectively by Baevski et al. (2020) and Algayres et al. (2022a). Wav2vec2.0 Base is a stack of 7 convolution layers and 12 transformer layers. The SSE is composed of a one GLU convolution layer (kernel size: 4, number of channels: 512, stride: 1), a transformer layer (attention heads: 4, size of attention matrices: 512 neurons, and FFN: 2048 neurons) and a final max-pooling layer along the time axis.

**LexEmb** is composed of two functions $L \circ q$. $L$ is a stack of five three-layers blocks each formed by a 1024-neurons fully connected layer, a layer

norm and a ReLU activation. $q$ is of a PCA and a collection of k-means that are trained on LL6k-clean. The PCA has $d = 24$ dimensions and the number of centroids for the first k-means is $K = 10$.

**Transformer** is identical to the one used in the original GSLM paper (Lakhotia et al., 2021). It contains 12 transformer layers with 16 heads, 1024-neuron attention matrices, and 4096-neurons FFN. On top of the transformer, the three parallel $h1,h2,h3$ functions are 1024-neurons fully connected layers. $L,h1,h2,h3$ and the transformer are trained on 32 GPUs, for 200k iterations on either the LibriSpeech or LL6k-clean. Each batch is composed of 64 audio sentences that are composed of 64 tokens. The learning rate is set at $5^{-4}$ with a warm-up of 5000 updates and polynomial decay. We use Adam optimizer with a weight decay of 0.1. A dropout of 0.1 is applied during training. The loss function is the NCE loss with a temperature of 0.1 and 500 negative samples.

**Sampling** is performed in a FAISS k-NN (Johnson et al., 2017) that contains all the lexical tokens segmented in the dev-clean and test-clean from the LibriSpeech (roughly 10 hours of speech). The number of nearest neighbours from which the next token is sampled is set to 1000.

**Speech generation model** is an encoder and a decoder that shares the same architecture: 4 transformer layers with 8 heads, 512-neurons attention matrices, and 3072-neurons FFN. It is trained on 32 GPUs, for 30k iterations on the LibriSpeech. Each batch is composed of four audio sentences that are at maximum 20 seconds long. The learning rate is set at $5^{-5}$ with a warm-up of $10^3$ updates and

polynomial decay. We use a dropout probability of 0.1 and Adam optimizer with a weight decay of 0.1. The Tacotron2.0 from Lakhotia et al. (2021); Kharitonov et al. (2022) was trained on LJ.

## A.4 Probing acoustic and lexical spaces

Figure 4 is a visualization of the acoustic and lexical representation learned by gold-tGSLM which echo a work on speech word embeddings from Chung and Glass (2018). All speech segments corresponding to real words in the LibriSpeech dev-clean set are indexed in k-NN graphs on their acoustic or lexical form. Each embedding is labelled with its true transcription. By searching for the nearest neighbors of a centre word (in red in the figure), we highlight in green the neighbours that we judged semantically related to the centre word. Figure 4 shows that an acoustic token has usually no semantically related neighbour other than ones with the same transcription. By contrast, lexical tokens have semantic and syntactic properties: 'London' is close to other cities and countries, 'blue' is close to colour names, beautiful is close to other positive adjectives, and 'chair' is close to 'desk' and 'table'. Nonetheless, it appears acoustic information has leaked from the acoustic tokens into the lexical tokens. For instance, the lexical neighbours of 'blue' are colours or shades that start with a 'b' and 'chest' appears in the neighbourhood of 'chair'.

## A.5 Estimation of memory consumption

To estimate the memory consumption of a transformer LM with $L = 16$ layers, a single attention head, a batch size of 1, and an embedding size $d = 1024$, let us write $x \in \mathbb{R}^{n \times d}$ a sentence of $n$ tokens represented with embeddings of size $d$. Using the formula expressed in Korthikanti et al. (2022), the number of activations to store in memory during backpropagation is approximately (buffers and negligible values being omitted) $\phi(L, n, d) = Lnd(34 + 5\frac{n}{d})$. In the LL6k-clean corpus, sentences are 60s-long on average with make $n = 1500$ for GSLM and $n = 300$ for 200ms-tGSLM. 200ms-tGSLM should expect a memory reduction by a factor of $\frac{\phi(16,1500,1024)}{\phi(16,300,1024)} \approx 5.83$ compared to GSLM. In practice, we observe a lower memory reduction ($\approx 4.76$) which can be explained by the additional parameters that are present in 200ms-tGSLM and not in GSLM, namely the $LexEmb$ function and three prediction heads.

## A.6 Inference time complexity

Taking the calculation of a forward cost from Pan et al. (2021), for a sequence length of size $n$, and a transformer of $L = 16$ layers and dimension $d = 1024$, a forward pass costs $(12nd + 2nd) * L$. This would be the cost of a forward pass in the GSLM model, but our tGSLM costs a little bit more with its LexEmb function and its sampling procedure. A forward through our LexEmb function costs $5nd^2$ (5 linear layers) and the sampling procedure costs $d * 100.000$ (we usually take 100k items for the k-NN search). Therefore, tGSLM cost $(12nd + 2nd)L + 5nd + d * 100000$. For a sequence of 1 second (therefore $n = 5$ for tGSLM and $n = 25$ for GSLM), by replacing those values in the former calculation, we find that a forward in tGSLM cost $1.1e6$ which is 5 times less costly than a forward in GSLM that would cost $5.2e6$. Therefore, even at inference time, our tGSLM should be much faster than GSLM to run.

**200ms-tGSLM examples**

**Generation at LJ-VERT**

What is it ask her mother i want to see you said mrs tumbled i want to tell you what you you said mr cockry you are no more chance than you know.

We have no desire to prevent to the astonishment of that person from the government who is not so far for receiving any property or relation to the world.

Her father in her son were under growth her father was just like a treasure man who was a devil and hazards beyond his words she was a very clearly.

We also see that it will be obliged to invite us to applyge them to observe such a thing is a base we must not set down that the

And although he was not equally successful to him he sought the pririate regularly observed his friends invent to him and presented him their own secret he had did

Because they were rested and although they could not expect to be obliged to regarded as a men of a gold and power they were not really unbusy

You see he is if i miss thing i think he is dead it said mrs carpenter rather smallly for anything if he is a total let she said

He remembered that great city which he tried to entertain in its pointedof view but he was very pleasant to him and could not bring whom away besides this

Having required a measure for a month before their distance of sixty years he appeared to be affected by any conditions of one state and have in no battle

Now the king's brothers came to him and brought him up and said i'll poor woman i woke it of you anything but i am brother and borrows my

**Generation at LS-VERT**

He turned his hands on the sale exposition and gave him to acy of old meal which wealth had never bore be a foreseenly large.

While waiting to him he wished that he would wait for himself into his mother's house and held light he was that that she might be able to look.

And perhaps i have nothing to say about what would you want to know i did i don't know i suppose you want to know what could call the

That's all i can't do insaid woman looking out of the croad toward him while i don't know such any end enging his hand you seem to see your

And having been described as the great activity of that which he was attempting all that if he now remar it for his purpose was intended with principles as

It was the time had been prepared for for that such was the place that when he was saing to his land she'd made up all though blood that he

It was just as willing to oppose the person who had been told of his chion he was now about to go to bank in the family to a

Having been in a moment' officially desirable to acquaint him with his reference with the glorious presence of his master's cabinet he did not return to a subject of

Yes i was said he but was a general service she began i could not file forhard seek i want to take my as andt understand the chance of

But afterwards they had gone to top what waking into the stone doors the weathering tight their habitation and the north were histor carryance and mr carb's face and

Table 8: Example generations of 200ms-tGSLM trained on LL6k-clean. These generations are selected from batches of sentences that have a VERT equal 0.113 (LS-VERT) or 0.189 (LJ-VERT).

**gold-tGSLM examples**

**Generation at LJ-VERT**

But you have been wanting to teach me all her life in the world of her own healthy health and she has her fathers abilities an your pride.

The old woman or that he would have learned all her life amongst the gods and teaching them in their father's studies and having been up the days.

It goes on until i know what i am doing while i am going away from my camp in the neighbourhood to morrow we you from the whole on my.

An elegant geographical character would you think it a deed or an excellent thing to do with the hold in the future won't you pick up a bit of an

The evening of the twenty fifth of november eighteen united eight he returned to his royal house an the hold of the hospital an the next evening he you

Guiding them in some ordinary way or buying them into cold or buying them with a copal spoon which should be thrown out of the souls of the bulkhead

Secies and germany each of them had undergone more than three thousand roubles and hour to saved a bit of jewelry from s odin share and the hardness and

Of the kavin and when to the door where she stood a few minutes later to reach the bottom of the harboured near the labyrinth where she reached her

It says the king listening the light of his bushy fingers an holding his pipe in his arms do not bother me any more about it you know more than

The investigation and on his returned to her fathers room he set down his gun at a hundred yards and the middle of the hall to learn the hut

**Generation at LS-VERT**

I can tell her that only one of my friends and loves do you think i would read her about this uttered it all the wicked said missus williams

She reached her big house and stood by the dora in turning to the king he said to her you will not marion me any more have you hear

To his voices and his broken heart screen with delight to henry smokeless who had entered into his dining room to limp him a mystic playful of his faintest

I shall not go without thee said heat pausing to her part a of good direction and fixing her ices upon her eyes with a distant cheeks to her.

Than time of missus esplanade visit her own house and china herself alone of theirlocal service and the frere settlements where built for thousand of the happiest teachers

Father and mother were all seated at boston waiting for the empty school at ostrog at nine o'clock a the knight of july evening a ninth jeanne annie eighteen

Minutes later he heard a bill calling against the young man who had denounce him his face became a melancholy shake in his astonishment what is it said george

A poor boy in has a good power for somebody's harm to be at heaven what could you to givewhat this seemed to him a hard proposition

Paper it was needless to be summoned to it by the princess and the girl became very much surprise and said about recovering the bicycle with her finger to

To touch it he s a gentle young ma'am and does not see any other foreign of mortality unconnected with her father who is afraid of his flesh to

Table 9: Example generations of gold-tGSLM, trained on LibriSpeech. These generations are selected from two batches of sentences that have a VERT equal 0.113 (LS-VERT) or 0.189 (LJ-VERT).

**GSLM examples**

**Generation at LJ-VERT**

They did time in the desert two or three hundred years afterward among them the castle was not my father and they were found in palestine by fire and they

Another excellent is descended a breath let the corp of a prisoner and a blow was begun the bell rang the gunsprang from the captain's paul and dropped into the

And then the passing future would have been too much but to waittill the end of the week and after a little time she had gone down to the palace

But he passed along entirely untouched and was still together so frightened in the morning he went to look out for some place with a barian laine and then he

The brast of the bravest of the entire youth and of many of the slaves of the counillllors or of every fine breed and of the princess of france has

He had not in the least delicate way of helping her but had helld her into a pretty soft and a passionate graceful manner he told her every day because

But that man did not fight in the second place no ne did pay the attention to poverty it is not tpossible to suspect that the man of the previous

But all the people had come to see me and had not seen me again and they felt as if i were again coming to see me and so little

And people stared at him for a moment as if they were dead but he had not told them of his destiny that he would do so and they had

And a cow calling up his pipe said that no sign of the procession was ever heard and that no punishment was made or judgment was made nor any other

**Generation at LS-VERT**

His proposals that being so poing doubful i should very much regard and alia in boa's addition to cloak the great morning had given me a plague off waivering she

Someone to found a brown line and dance spent a moment over the vessel all saw the fair young chinese yard and dry he would waved dances in bubbles from

All cathics that are not due to f co notion or naturalist that is intentionble but if there is a personality of faith in them who was intentupon for seeing

The reef made the partets at the corner of a platform with them rose and ground on the floor of the lobby and of chapter fourteen two thousand se of

As if sudden impulse were convinced of their usual impulses and a strong exercise upon them or rather in their progress to bring their education to the reduction of manly

And rushing off from the cold winds in the west in the silence of the rock the cherry wavering soft quietness of people makes breath so cheap in a course

He had been burden with visitor and had petched his old preserance for death and mary the intamminable enterprising scenes caused by constantiis this trumpetts und drrawn courage and he

Great worked done an artificial lines of bounding is below the had an arm as it were it lookedfted itself and everything was so exquisite that the site was hard

To jew knew that they had been driven a doctor adreadful mattering to you the young girl whom eyed by a relatives ever since daily matters while a week before

Evenings in a ball volume whose close ways were rotted in whther cuts that fiddler devilalonsome his wife soldiers were harassing a women with deafferenren american last grading under fair

Table 10: Example generations of GSLM, trained on LL6k. These generations are selected from two batches of sentences that have a VERT equal 0.113 (LS-VERT) or 0.189 (LJ-VERT).

|            | WUGGY↑ | SBLIMP↑ |
|------------|--------|---------|
| GSLM       | 70.36  | **56.31** |
| 200ms-tGSLM | 68.53 | 55.31   |
| BC-30k     | **71.32** | 55.08 |
| BC-2k      | 69.00  | 54.44   |

Table 11: sWUGGY and sBLiMP GSLM and 200ms-tGSLM and the large units

# B Clustering SSEs

## B.1 The problem of clustering large units

In the introduction, we argued that the clustering of a large collection of word-size speech fragments is a daunting challenge. The first difficulty is the very large number of word types in a corpus. For instance, there are ≈4.5k word types in the 10-hour-long Buckeye corpus (Pitt et al., 2005), and ≈90k in the 960-hour-long LibriSpeech (Panayotov et al., 2015). Most clustering algorithms (k-means, hierarchical-k-means, Spectral, Chinese Whisper (Biemann, 2006), Dirichlet Process-means (Kulis and Jordan, 2011), Brown (Brown et al., 1992)), require in input either this unknown number of clusters or a threshold parameter that controls the number of clusters[9]. By misestimating the number of clusters, we introduce errors that can mislead the downstream LM. The second problem is the highly skewed distribution of the frequency of word types. This phenomenon, known as Zipf's Law (Zipf, 1949), states that, in text, there is a linear relation between the *log*-rank of word types and their *log*-frequencies. In practice, it means that, in a text, most word tokens are *hapaxes* (i.e. they have a word type that appears only once), and a small proportion of word types account for most tokens. Therefore, even if the number of word types could be correctly estimated, a clustering algorithm would have to produce many singleton clusters, which is a hard task for clustering models, especially for k-means that tend to create equal-size clusters (Wu, 2012). For those reasons, a clustering algorithm is likely to produce non-recoverable errors that will negatively impact the downstream spoken LM.

## B.2 Attempt at clustering SSEs

Here are some attempts and results at the task of clustering speech fragments. First, we retrieved the gold-standard word segmentation of the LibriSpeech corpus and embedded all word tokens with the SSE model from Algayres et al. (2022a). Then, we clustered with k-means and hierarchical-k-means all the SSEs in the LibriSpeech into $K$ classes. To simplify, we set $K$ to the true number of word types. Finally, we trained a transformer LM to predict the next discrete clusters. We observed that the training loss decreased but not the validation loss. The reason for this failure was simple: most of the clusters found by k-means or hierarchical-k-means that were in the validation set were not present in the training set. These results show that those baseline clustering algorithms are not suited to the task of spoken language modelling.

Regarding more elaborate clustering methods, Elkahky et al. (2023) has used a combination of HuBERT units, smoothing technics, Byte-Pair-Encoding (BPE) and Brown clustering (BC) (Brown et al., 1992) with either 30k clusters or 2k clusters. The result is a discretisation of speech with word-size units. We have proved in Elkahky et al. (2023) that these units can be used to train a HuBERT model and improves its downstream ASR performances. Here, we discretised the Libri-Light 6k clean (Rivière and Dupoux, 2021) with these large units using either 30k and 2k clusters and gave them in input to a transformer LM trained to predict the next discrete units. We report in Table 11 sWUGGY and sBLIMP scores of GSLM, 200ms-tGSLM and the large units (called BC-30k and BC-2k). The scores show that these units perform equally well to GSLM and 200ms-tGSLM. These results show that even the most elaborate methods of clustering do not bring better results than our method to adapt spoken LM to continuous inputs.

---

[9]Some unsupervised methods to find the number of clusters exist, like the 'elbow methods' in k-means, but these methods are quite noisy and hard to apply when the number of classes is that large.