# OpenReview forum: "Generative Spoken Language Model based on continuous word-sized audio tokens"
_EMNLP/2023/Conference — EMNLP 2023 Main_

### Official Review · Reviewer_BKbZ · 2023-07-26

**Soundness:** 4

**Excitement:**

4: Strong: This paper deepens the understanding of some phenomenon or lowers the barriers to an existing research direction.

**Paper Topic And Main Contributions:**

This paper proposes tGSLM, which is a generative spoken language model with features aligned with lexicon encodings. The advantage of this method is that it operates at a scale similar to phonetic/lexical units so that it achieves high efficiency while maintaining the same performance as the fine-grained GSLM.

**Questions For The Authors:**

1. How do you get the corresponding Lex tokens for a corresponding 200ms window exactly? Do the sequence of lexical tokens and the sequence of acoustic tokens have the same lengths?
2. Have you tried any other lengths rather than 200ms?

**Reasons To Accept:**

1. The alignment between acoustic encoding and lexical encoding to help GSLM is innovative.
2. Experiments are thorough and in detail.

**Reasons To Reject:**

1. As the method involves a lot of specific knowledge and dedicated components, it is a bit hard to follow the description of the system. In particular, I was unclear about how the upsampling to HuBERT units was done. Also, see questions for the authors below.

**Reproducibility:**

4: Could mostly reproduce the results, but there may be some variation because of sample variance or minor variations in their interpretation of the protocol or method.

**Reviewer Confidence:**

4: Quite sure. I tried to check the important points carefully. It's unlikely, though conceivable, that I missed something that should affect my ratings.

---

> ### Author Rebuttal · Authors · 2023-08-25
>
> We thank the reviewer very much for their time and effort into reviewing our paper. We understand that this paper is quite long and complex with many new components. Here are the answer to your questions.
>
> "I was unclear about how the upsampling to HuBERT units was done"
>
> Indeed our section 3.2.2 that explains the decoding to HuBERT units is not very clearly formulated. In this section, we wished to explain that the HuBERT units are obtained with a transformer encoder-decoder that takes acoustic tokens in input and outputs HuBERT units. In appendix A.3 we explain how we train this model on the librispeech corpus.
>
> "How do you get the corresponding Lex tokens for a corresponding 200ms window exactly? Do the sequence of lexical tokens and the sequence of acoustic tokens have the same lengths?"
>
> Regarding this, we will try to make section 3.1.1 and 3.1.2 clearer. These two sections explain how acoustic tokens are built and how each one of them is mapped to a single lexical token with the LexEmb function. This one to one mapping is maybe more visible on figure 1 and 2.
>
> "Have you tried any other lengths rather than 200ms?"
>
> Indeed we have tried, an ablation on the duration of units is provided in appendix at table 4.
>
> As show in this rebuttal, for each of your questions, there is a dedicated section that answer it. We wish that these answers are satisfactory to you and that you may consider giving this paper a higher grade. Thank you very much for all your comments that will make this paper much clearer and much easier to read.

---

### Official Review · Reviewer_esLy · 2023-08-03

**Soundness:** 4

**Excitement:**

4: Strong: This paper deepens the understanding of some phenomenon or lowers the barriers to an existing research direction.

**Paper Topic And Main Contributions:**

The paper proposes a method to transform and concatenate frames extracted by wav2vec 2.0 into 200ms vector-based acoustic tokens. The vector-based acoustic tokens can be applied for training a 200ms-tGSLM speech based language model. Performance of this  200ms-tGSLM is comparable to conventional GSLM based on 240ms frames. The proposed method has the potential to improve speech LLM modelling, in terms of computation efficiency and representation ability.

**Questions For The Authors:**

A. Section 3.2.2: For my understanding, the generated acoustic token vectors are converted into Hubert before applying to Tacotron2.0 vocoder. I understand that the authors try to reuse the available vocoder for comparison. However, directly generating time domain signals from the 200ms acoustic tokens with techniques such as GAN should be feasible, and may achieve better results. Have the authors tried the experiments to generate speech directly from the 200ms tokens?

B.Section 3.1.2: The quantisation parameter $q$ is pre-computed before training. How to determine the PCA size and the K-mean cluster size? Is it possible to replace this pre-computed parameter with other quantisation such as VQ such that this bottle net layer can be trained end-to-end?

**Reasons To Accept:**

The significant longer sequence length is a challenge of applying NLP sequence-to-sequence technique to speech. There are a lot of researches trying to shorten speech sequences for effective computation and better performance. This paper provides another perspective by proposing a method to generate longer fixed size acoustic tokens of 200ms which is much close to duration of sub-word units. The length the speech sequence is now comparable to the sequence with sub-word tokens, which has the potential for improving speech LLM training. While the choice of 200ms is somehow heuristic, the duration aligns well with average syllable length of most languages. The work includes experiments of different tasks and the results are encouraging.

**Reasons To Reject:**

The performance of 200ms-tGSLM is not necessary better than conventional 40ms based GSLM, except the MMOS for generated speech. However, I do not think that this is  a strong reason to reject this paper.

**Reproducibility:**

4: Could mostly reproduce the results, but there may be some variation because of sample variance or minor variations in their interpretation of the protocol or method.

**Reviewer Confidence:**

3: Pretty sure, but there's a chance I missed something. Although I have a good feel for this area in general, I did not carefully check the paper's details, e.g., the math, experimental design, or novelty.

---

> ### Author Rebuttal · Authors · 2023-08-25
>
> We thank the reviewer very much for their time and effort into reviewing our paper. Overall we agree that our implementation choices are debatable, and that a tGSLM version 2 should follow your two propositions. Here are the responses to your questions and comments.
>
> “For my understanding, the generated acoustic token vectors are converted into Hubert before applying to Tacotron2.0 vocoder. I understand that the authors try to reuse the available vocoder for comparison. However, directly generating time domain signals from the 200ms acoustic tokens with techniques such as GAN should be feasible, and may achieve better results. Have the authors tried the experiments to generate speech directly from the 200ms tokens?”
>
> This is indeed the main weakness of our paper. The model would be much stronger with a specific decoder that would map acoustic tokens to speech. We are thinking to do it with a Hifi-GAN model that we would adapt to take acoustic tokens in input instead of the hubert units.
>
> “The quantisation parameter is pre-computed before training. How to determine the PCA size and the K-mean cluster size? Is it possible to replace this pre-computed parameter with other quantisation such as VQ such that this bottle net layer can be trained end-to-end?”
>
> Indeed the quantization is precomputed before training. In appendix section A.2.1 we give a little bit of details on the PCA and k-means parameters. Basically, we have gridsearched the PCA dimension and k-means clusters to maximize the zero-shot ABX metrics on a smaller dataset (the librispeech) than the one used for training (the LL6k clean). We agree that it would be a nicer solution to use learned VQ with a gumble softmax as the authors of Wav2vec2.0 did. We decided to go for PCA and kmeans to make the model easier and lighter to train (in wav2vec2.0 they had to incorporate a diversity loss with a weighting parameter to make the VQ quantisation work)
>
> Thank you very much for all your comments that will make this paper much clearer and much easier to read.

---

### Official Review · Reviewer_Uy5K · 2023-08-10

**Soundness:** 4

**Excitement:**

3: Ambivalent: It has merits (e.g., it reports state-of-the-art results, the idea is nice), but there are key weaknesses (e.g., it describes incremental work), and it can significantly benefit from another round of revision. However, I won't object to accepting it if my co-reviewers champion it.

**Missing References:**

In your introduction, you mention "tokenizer representing word or subword units" (l60) without mentioning examples of such method. If think the comparison with your method is very relevant. It could then be nice (if you have enough space) to mention and cite the most popular methods currently used to build vocabularies for text, that the reader can relate to:

* [BPE](https://www.derczynski.com/papers/archive/BPE_Gage.pdf) and [its application to natural language](https://aclanthology.org/P16-1162/)
* [WordPiece](https://arxiv.org/abs/1609.08144v2)
* [Unigram](https://aclanthology.org/P18-1007/)

NCE loss introduction: [Noise-contrastive estimation: A new estimation principle for unnormalized statistical models](http://proceedings.mlr.press/v9/gutmann10a.html)

**Paper Topic And Main Contributions:**

The paper treats of "audio" language modeling, that is language represented as audio spoken modality. This task is tackled by training a language model (LM) on speech data represented as discrete tokens. Such model work similarly to text LMs, with a fixed-size vocabulary, cross-entropy loss during training and multinomial sampling during inference, and can be used jointly with vocoder models to autoregressively generate speech and audio.
In previous works, the vocabulary is built from sets of learned speech sequence embeddings (SSE) of sequences of speech. These sequences of speech are often short, between 20 to 40ms, and may however not be representative of any relevant information (i.e. when shorter than a phoneme or syllable). An other drawback is the extensive token sequence length created by that such method, which is a known bottleneck of Transformers / attention mechanism.
In this paper, the authors propose to use tokens of longer speech sequences, which would reduce the token sequence length and in turn reduce the overall complexity / memory usage, while having more semantically rich audio embeddings. Doing so will however create a very large number of possible SSEs / tokens, making having a finite and well balanced vocabulary intractable (called clustering problem). The authors propose then to directly work in a continuous space, that is not using a finite vocabulary but directly feed the model with continuous vectors. The model cannot directly be fed the SEEs, as "as two speech segments may sound the same, i.e. be close in the acoustic space", so an additional function, learned with the model ("lexical embedder") mapping the SEEs to input embeddings (called here "lexical tokens"), is designed. And as there is no finite vocabulary, the model cannot be trained with the cross-entropy loss commonly used. The authors resort to use a contrastive learning loss. Another modification is made on the output of the model: the authors used three separate output heads (fully connecter layers) to predict the next three tokens. This modification allows to "extend" the scope of prediction of the model, which would be fairly limited by autoregressively sampling one token representing a piece of a word after another.
Compared to the [GSLM](https://aclanthology.org/2021.tacl-1.79/) baseline, the method proposed by the authors allows to significantly reduce the memory usage (by reducing the input sequence lengths) while having results of better quality (measured by perplexity and zero-shot metrics, along with human preferences) and interpretable embeddings.

**Questions For The Authors:**

**A.** I think the "lexical tokens" and "acoustic tokens" namings are confusing at first read, and not really accurate. In the literature, a token is a discrete element, from a finite ensemble. I would recommend to rename them to something that might indicate that these elements are actually continuous / vectors from a continuous space. What is your opinion on this?

**B.** Did you made some benchmarks on the generation speed? I think it could be express in tokens/sec and sec_of_audio/sec. As your model now has additional modules (namely LexEmb, multiple output heads, using FAISS search), I assume that the generation speed gain is not linear to the token sequence reduction;

**C.** Maybe a suggestion for future research: what do you think about conditioning the output heads on the results of the previous one? By that I mean that $\mathbf{y}\_i = h(\cdot)$ could be conditioned on $\mathbf{y}\_{i-1}$, e.g. $\mathbf{y}\_i = h(\mathbf{x}, \mathbf{y}\_{i-1})$. Some models applied the same method of yours for symbolic music modeling ([Compound Word](https://ojs.aaai.org/index.php/AAAI/article/view/16091), [MusicBERT](https://aclanthology.org/2021.findings-acl.70/)). In practice, generating several distinct elements at the same time, unconditionally although they are meant to be complementary / tied, does not work very well for causal generation (as GANS / Diffusion / any non-autoregressive model). Thus conditioning each output on the very last predicted is almost always guaranteed to give more coherent results. [Compound Word](https://ojs.aaai.org/index.php/AAAI/article/view/16091) does it partially.

**Reasons To Accept:**

* The overall goal is very well motivated. Tackling audio modeling as a sequential problem is a very good direction offering a lot of flexibility in real-condition usages. I believe there is a lot of progress to be made, with significant impact, on the representation / tokenization of audio;
* The implementation choices of tGSLM are well made and well justified (e.g. 200ms segmentation). Tackling this problem required a fair amount of model adaptations (LexEmb, using contrastive loss, FAISS for inference...) that the authors cleverly overcame;
* The tasks and set of metrics are well chosen;
* The interpretation of the "lexical token" embeddings is valuable and empirically shows that the method works;
* The results show that LMs can generate speech using word-size frames directly fed as continuous embeddings;
* The paper is globally well written and structured;
* The references are very well documented and completed. Only one or two has missing URLs.

**Reasons To Reject:**

* Although the experimental setup is well made (tasks + metrics), I think the paper can really benefits from one or two more baselines, even though this kind of audio representation is fairly new and that you answer well to the main goal of the paper that you set (i.e. generate speech using word-size audio embeddings). The authors could add tokenizations from [AudioLM](https://arxiv.org/abs/2209.03143) or [BEATs](https://arxiv.org/abs/2212.09058). Eventually a baseline without $LexEmb$, that could highlight its necessity. I think the reader would benefit to see how tGSLM compares to these baselines.
* There is no information about generation speed at inference. This information is IMO as well important as the memory usage. It as a decisive point when designing a system / product. Here, considering the additional stack (FAISS) compared to models using finite vocabularies, the reader could want to know id it is worth to spend the time needed to implement it.
* **(no big deal)** The memory gain estimation in A.5 is valuable. But it would be great to also consider the FAISS memory consumption for inference, and give concrete examples with numbers, e.g. how $N$ pairs of SEE / lexical token translate into memory consumption. This $N$ could be a limitation depending on the hardware being used. Also as $N$ is correlated to the lengths of the speech sequences, it could raise questions when using such model in real conditions.

**Reproducibility:**

5: Could easily reproduce the results.

**Reviewer Confidence:**

3: Pretty sure, but there's a chance I missed something. Although I have a good feel for this area in general, I did not carefully check the paper's details, e.g., the math, experimental design, or novelty.

**Typos Grammar Style And Presentation Improvements:**

You should consider converting your figures to vector format, this will provide the reader a better quality of visualization. Assuming Fig. 3 and 4 are generated with pyplot, you just have to save them as pdf.

Rename "lexical tokens" and "acoustic tokens" (see reasons to reject);

Fig. 1: you could change the contour of the "Lex Emb" contours in order to better distinguish methods/functions from models, change "Transformer" to "LM" (more general), add the signification of "FC" somewhere in the main text;

l244: "there is often a linear FC layer before the transformer" --> this is not really accurate, it isn't a fully-connected layer. You should rephrase with something like "a learned embedding matrix acting as a lookup table";

l255: "measured" --> did you mean studied ?

l267: precise acoustic tokens;

l322: add signification of NCE + cite [Gutmann et al](http://proceedings.mlr.press/v9/gutmann10a.html);

l322 footnote (3): It would be good to include in the main text that L2 reconstruction is applicable, but did not get you good results in practice and include them in appendix;

l356 / subsection 3.2.1: You could mention that you are using FAISS (I assumed it was when looking at fig 2 but had to look for appendix to find the answer);

l1080 footnote: "those" --> "these" + missing dot;

l1355: add the signification of "BPE"

---

> ### Author Rebuttal · Authors · 2023-08-28
>
> We thank the reviewer very much for their time and effort into reviewing our paper. This review is impressively complete and thoughtful and will definitely help us a lot in improving the paper. Here are the responses to your questions and comments.
>
> "The authors could add tokenizations from AudioLM or BEATs."
>
> Indeed, our paper is missing a comparison with AudioLM  (it did not exist yet when we started to work on tgslm) which has a similar (or slightly higher) semantic consistency than the original GSLM.  Yet, adding audioLM to our paper would not change the main takeaway which is that we get similar performances with continuous 200ms GSLM compared to discrete 40ms GSLM.
>
> "There is no information about generation speed at inference [and] it would be great to also consider the FAISS memory consumption for inference, and give concrete examples with numbers, e.g. how pairs of SEE / lexical token translate into memory consumption. This could be a limitation depending on the hardware being used. Also as is correlated to the lengths of the speech sequences, it could raise questions when using such model in real conditions. [...] Did you made some benchmarks on the generation speed? I think it could be express in tokens/sec and sec_of_audio/sec. As your model now has additional modules (namely LexEmb, multiple output heads, using FAISS search), I assume that the generation speed gain is not linear to the token sequence reduction"
>
> Indeed the analysis on inference time is absent from the paper and could be a nice addition. Here is a little paragraph that we may add to the camera-ready paper.
>
> Taking our calculation from A.5, for a sequence length of size n, and a transformer of L layers and dimension d, a forward pass costs
>
> (12nd²+2n²d)*L FLOP
>
>  This would be the cost of a forward pass in the GSLM model, but our tGSLM costs a little bit more with its LexEmb function and its sampling procedure.
> A forward through our LexEmb function cost 5*n*d² (5 linear layers) and the sampling procedure cost d*100.000 (we usually take 100k item for the k-NN search). Therefore, tGSLM cost
>
> (12nd²+2n²d)L + 5nd²+d*100000 FLOP
>
> For a transformer of d=1024 and L=16, and a sequence of 1 second (therefore n=5 for tGSLM and n=25 for GSLM), by replacing those values in the former calculation, we find that a forward in tGSLM cost 1.1e6 FLOP which is 5 times less costly than a forward in GSLM that would cost 5.2e6 FLOP.
> Therefore, even at inference time, our tGSLM should be much faster than GSLM to run.
>
> "I think the "lexical tokens" and "acoustic tokens" namings are confusing at first read, and not really accurate. In the literature, a token is a discrete element, from a finite ensemble. I would recommend renaming them to something that might indicate that these elements are actually continuous / vectors from a continuous space. What is your opinion on this?"
>
> I agree with you, we could maybe call them acoustic embeddings and lexical embeddings.
>
> "Maybe a suggestion for future research: what do you think about conditioning the output heads on the results of the previous one?"
>
> We did think about this and I might be wrong here but I think it would actually impair performances. Let me try to explain why. The main reason we use several heads is to prevent the causal LM to make prediction by relying on coarticulation (local acoustic information that leaks from one token to the next). This problem emerges because we have continuous tokens (and not discrete ones) that can carry low-level information. Consequently, the token at time t, is a little bit coarticulated with the token at t+1 (both at the acoustic and lexical level), but not coarticulated with tokens at t+2 or t+3 (because coarticulation is typically shorter than a phoneme or 80ms). By doing the three head predictions, the LM cannot rely on coarticulation too much (it actually can for head 1, but not for head 2 nor 3).
> To come back to your proposition, if we use conditionning on the last result, we basically enable the head to use coarticulation by giving it the last predicted token. But I might be wrong and may it would improve performances as you propose.
>
> Thank you very much for all your comments, corrections and suggestions, that will make this paper much clearer and much easier to read.

---

### Meta-Review · Area_Chair_BFkt · 2023-09-19

**Recommendation:** 5

**Metareview:**

This paper proposes to replace 20ms or 40ms-long discrete units used for generative spoken language modelling with 200ms-long continuous word-sized audio tokens. The paper explains the architectural choices needed to achieve this goal and reports that the model's performance is on par with the standard discrete-unit-based language model while being more memory efficient.

The reviewers praised clear writing, good motivation, explanation of the architectural choices, and sound experimental design. The reviewers raised several questions, which the authors answered during the rebuttal. The only comment that needs to be addressed in the camera-ready version is empirical inference speed measures, such as sec of generated audio per sec of inference.

---

### Decision · Program_Chairs · 2023-10-07

**Decision:**

Accept-Main

**Comment:**

This paper proposes to replace 20ms or 40ms-long discrete units used for generative spoken language modelling with 200ms-long continuous word-sized audio tokens. The paper explains the architectural choices needed to achieve this goal and reports that the model's performance is on par with the standard discrete-unit-based language model while being more memory efficient.

The reviewers praised clear writing, good motivation, explanation of the architectural choices, and sound experimental design. The reviewers raised several questions, which the authors answered during the rebuttal. The only comment that needs to be addressed in the camera-ready version is empirical inference speed measures, such as sec of generated audio per sec of inference.